# Structural and functional characterization of a DARPin which inhibits Ras nucleotide exchange

Sandrine Guillard[1], Paulina Kolasinska-Zwierz[1], Judit Debreczeni[2], Jason Breed[2], Jing Zhang[3], Nicolas Bery[3], Rose Marwood[1], Jon Tart[2], Ross Overman[2], Pawel Stocki[1], Bina Mistry[1], Christopher Phillips[2], Terence Rabbitts[3], Ronald Jackson[1] & Ralph Minter[1]

Ras mutations are the oncogenic drivers of many human cancers and yet there are still no approved Ras-targeted cancer therapies. Inhibition of Ras nucleotide exchange is a promising new approach but better understanding of this mechanism of action is needed. Here we describe an antibody mimetic, DARPin K27, which inhibits nucleotide exchange of Ras. K27 binds preferentially to the inactive Ras GDP form with a $K_d$ of 4 nM and structural studies support its selectivity for inactive Ras. Intracellular expression of K27 significantly reduces the amount of active Ras, inhibits downstream signalling, in particular the levels of phosphorylated ERK, and slows the growth in soft agar of HCT116 cells. K27 is a potent, non-covalent inhibitor of nucleotide exchange, showing consistent effects across different isoforms of Ras, including wild-type and oncogenic mutant forms.

[1] Antibody Discovery and Protein Engineering, MedImmune, Milstein Building, Granta Park, Cambridge CB21 6GH, UK. [2] Discovery Sciences, Innovative Medicines and Early Development, AstraZeneca, Darwin Building, Cambridge Science Park, Milton Road, Cambridge CB4 0WG, UK. [3] MRC Molecular Haematology Unit, Weatherall Institute of Molecular Medicine, John Radcliffe Hospital, University of Oxford, Oxford OX3 9DS, UK. Correspondence and requests for materials should be addressed to R. Minter (email: minterr@medimmune.com).

Wild-type Ras GTPases cycle between an active, guanosine 5′-triphosphate (GTP)-bound state and an inactive, guanosine 5′-diphosphate (GDP)-bound state[1–4]. This is mediated by nucleotide exchange factors, such as Son of Sevenless (Sos), which catalyse the exchange of GDP for GTP and GTPase activating proteins, which potentiate the weak intrinsic GTPase activity of the Ras protein[5]. Molecular activation triggered by extracellular stimuli such as epidermal growth factor (EGF) increases intracellular Ras-GTP levels and consequent interaction with downstream effector proteins. Cancer causing mutations impair the GTPase activity of Ras, leading it to accumulate in the activated state[6–8]. Cancer cells expressing mutant Ras are dependent on Ras for proliferation and survival. The RAS gene family, comprising HRAS, KRAS and NRAS, is the most frequently mutated oncogene family found in human tumours, especially pancreatic, lung and colon cancers[3,4,9]. Despite the prevalence of RAS mutations in cancer, no therapies that directly target the oncoprotein are currently available in the clinic.

The Ras molecule contains a number of sites essential to its function. The P-loop (residues G10-S17) is responsible for phosphate binding, whereas the Switch I (D30-D38) and Switch II regions (G60-E76) are critical for interactions with guanosine nucleotide exchange factors and effector proteins[3,10,11]. Three approaches demonstrated to directly modulate Ras function within the cell are: increasing hydrolysis rates of mutant Ras[12,13], blocking the interaction between active Ras and downstream effectors such as Raf, or reducing the amount of active Ras within the cell by inhibiting nucleotide exchange. The principle of inhibition of the Ras/Raf interaction has been supported by studies with the small molecule Ras-mimetic rigosertib[14], currently in phase III clinical trials for myelodysplastic syndrome, and a single domain antibody iDab6, which binds to the switch 1 site[15–17]. When expressed within the cell, iDab6 inhibits signalling downstream of Ras, can revert the transformed phenotype and, in mouse models, reduces growth of Ras-driven tumours. In contrast, the effects of blocking nucleotide exchange on Ras function in cells are thought to be restricted by the low GTPase activity of Ras mutants. Peptides and small molecules that bind directly non-covalently to Ras and inhibit interactions with Sos have been reported[18–20] together with initial effects on pERK signalling. Recently, however, a potent small molecule ARS-853 has been described that specifically inhibits nucleotide exchange by G12C mutants of K-Ras by covalent modification at the cysteine residue. ARS-853 reduces the amount of active Ras, increases cleavage of PARP and triggers apoptosis[21,22]. The potency of ARS-853 in reducing active Ras levels by >90% and substantially inhibiting both the ERK and AKT effector pathways has led some in the Ras field to re-evaluate the potential of nucleotide exchange inhibition, particularly for Ras mutants with intrinsically higher rates of GTP hydrolysis[23].

To increase understanding of the effects of non-covalent inhibition of nucleotide exchange on multiple isoforms and mutant forms of Ras, we have isolated a pan-reactive antibody mimetic, Designed Ankyrin Repeat Protein (DARPin) K27, which potently inhibits nucleotide exchange, and also a comparator, DARPin K55, which inhibits the Ras/Raf interaction. DARPins are small (< 20 kDa), antibody mimetic proteins which can bind to antigens with high affinity and specificity but, unlike antibodies, are particularly suited for targeting intracellular proteins since they lack cysteines and are highly stable allowing functional binding against intracellular targets[24,25]. Intracellular expression of the DARPins reported in this paper has allowed investigation of the effects of a potent inhibitor of nucleotide exchange at Ras on levels of active Ras and Ras function.

## Results

**Isolation of DARPins specific for Ras.** DARPins specific for Ras were isolated from phage display libraries. Selections were performed for two rounds on 250 nM and one round on 25 nM K-Ras G12V. DARPins from phage display were screened by performing immunoassays on K-Ras G12V and specific binders were subsequently sequenced to identify the unique DARPins. To identify inhibitors of Ras, purified DARPins were screened in a two-step assay, comprising a first step of Sos-mediated nucleotide exchange and a second step of Ras/Raf binding ('Ras biochemical coupled assay'; Supplementary Fig. 1). DARPins that were active in this screen could potentially inhibit at either stage of the assay but were further characterized for nucleotide exchange and Ras/Raf inhibition respectively. Some DARPins inhibited exchange of N-methylanthraniloyl (MANT) derivatives of guanosine nucleotides and were designated as nucleotide exchange inhibitors, while others directly blocked the interaction between active, GTP-loaded Ras and Raf and were designated Ras/Raf inhibitors. The amino acid sequences and inhibitory mechanisms of six active DARPins are summarized in Supplementary Fig. 2. The most potent DARPin in each class was selected for further analysis. DARPin K27 was selected as the nucleotide exchange inhibitor and DARPin K55 as the Ras/Raf inhibitor.

**Biochemical characterization of DARPins K27 and K55.** DARPin K27 and K55 binding kinetics on K-Ras, in both the GDP and GTPγS forms, were studied using bio-layer interferometry. It is worth highlighting at this point that both wild type (wt) and G12V K-Ras were used for biochemical characterization since we observed no mutation-specific binding for either DARPin, and subsequently found equivalent binding to all isoforms and mutants. DARPin K27 bound strongly to K-Ras wt in the GDP form with a dissociation constant of 3.9 nM (Table 1, Fig. 1). An accurate dissociation constant could not be determined for DARPin K27 for K-Ras wt in the GTPγS form since dissociation was biphasic, presumably reflecting slow dissociation from the contaminating GDP form, present due to hydrolysis, and faster dissociation from K-Ras wt GTPγS (Fig. 1). The rapid dissociation rates of DARPin K27 from Ras GTPγS at concentrations above 1 μM were indicative of a $K_d$ of DARPin K27 for the GTPγS form well above 1 μM. In contrast, DARPin K55 did not bind to K-Ras wt GDP but clearly bound to K-Ras wt loaded with GTPγS with a dissociation constant of 167 nM (Table 1, Fig. 1).

**Table 1 | Activity of DARPins K27 and K55 in biochemical assays defines their mechanism of inhibition.**

| DARPin | K-Ras GDP $K_d$, nM | K-Ras GTPγS $K_d$, nM | K-Ras biochemical coupled assay IC$_{50}$, nM | K-Ras/Raf assay IC$_{50}$, nM |
|---|---|---|---|---|
| K27 | 3.9 | ≫1,000* | 2.4 | Incomplete inhibition† |
| K55 | No binding | 167 | 67 | 120 |

*$K_d$ could not be accurately determined due to a biphasic dissociation phase, but was estimated to be well above 1,000 nM.
†Inhibition was incomplete at the maximum K27 concentration of 23.6 μM.

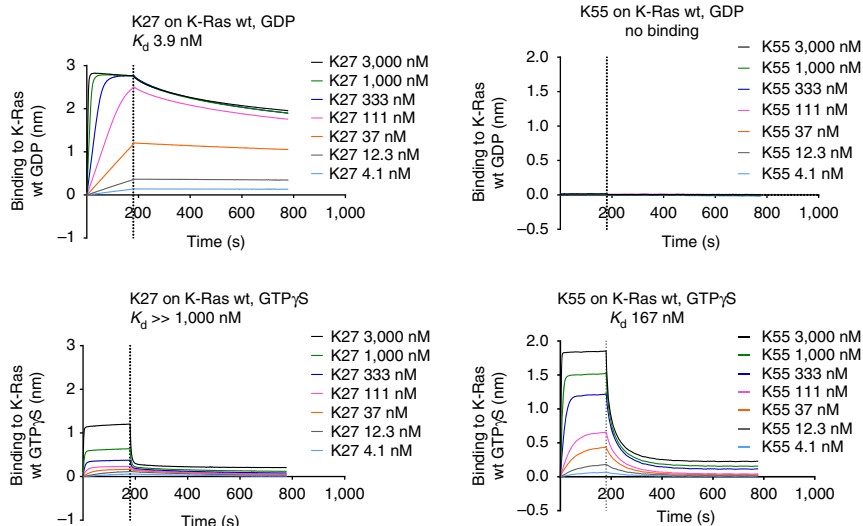

**Figure 1 | Bio-layer interferometry of DARPins K27 and K55 binding to K-Ras wild type loaded with GDP or GTPγS.** Bio-layer interferometry was used to investigate DARPin binding to biotinylated K-Ras wild type loaded with different nucleotides, immobilized on a streptavidin biosensor. The vertical dotted line indicates the transition from association to dissociation phases. Calculated $K_d$ values for each experiment are shown. Replicates are not included in this data but the experiment was repeated twice with similar results.

| Table 2 | Data collection and refinement statistics. | |
| --- | --- | --- |
| | **K27** | **K55** |
| *Data collection** | | |
| Space group | P6⁵ | H3 |
| Cell dimensions | | |
| *a, b, c* (Å) | 197.14, 197.14, 84.61 | 112.83, 66.77 |
| α, β, γ (°) | 90, 90, 120 | 90, 90, 120 |
| Resolution (Å)** | 3.22–98.57 (3.22–3.30) | 2.19–56.41 (2.19–2.25) |
| Number of reflections | 309,691 (22,248) | 71,531 (2,298) |
| Number of unique reflections | 30,646 (2258) | 15,629 (903) |
| $R_{merge}$ | 0.362 (0.965) | 0.074 (0.716) |
| CC(1/2) | 0.988 (0.562) | 0.998 (0.477) |
| $I/\sigma I$ | 7.8 (1.8) | 12.8 (1.3) |
| Completeness (%) | 100 (100) | 95.7 (74.4) |
| Redundancy | 10.1 (909) | 4.6 (2.5) |
| | | |
| *Refinement* | | |
| Resolution (Å) | 3.22–98.57 (3.22–3.30) | 2.19–56.41 (2.19–2.24) |
| No. of reflections | 29,155 (2124) | 14,863 (868) |
| $R_{work}/R_{free}$ | 0.201/0.235 | 0.185/0.231 |
| No. atoms | | |
| Protein | 9,731 | 2,323 |
| Ligand/ion | 55.66 | 36.93 |
| Water | 41.29 | 38.5 |
| *B-factors* | | |
| Protein | 75.9 | 42.9 |
| Ligand/ion | 55.66 | 36.93 |
| Water | 41.29 | 38.5 |
| R.m.s. deviations | | |
| Bond lengths (Å) | 0.015 | 0.013 |
| Bond angles (°) | 1.956 | 1.618 |

*One crystal was used in both experiments. **Values in parentheses are for highest-resolution shell.

As in the initial screening, DARPins K27 and K55 were both active in the Ras biochemical coupled assay, which encompassed the two steps of nucleotide exchange and Ras/Raf inhibition, yielding $IC_{50}$ values of 2.4 and 67 nM respectively (Table 1, Supplementary Fig. 3). In order to better understand the relative contribution of nucleotide exchange inhibition versus direct Ras/Raf inhibition for each DARPin, the Ras biochemical coupled assay was also performed as a pure Ras/Raf inhibition assay, using K-Ras pre-loaded with GTPγS. DARPin K55 inhibited with an $IC_{50}$ of 120 nM (Table 1), suggesting that the potency of this DARPin is primarily attributable to direct blocking of the Ras/Raf interaction. This is also in agreement with the strong preference of DARPin K55 for GTP-loaded, active Ras in the bio-layer interferometry assay. For DARPin K27, inhibition of the interaction of Raf with GTPγS loaded Ras was incomplete at the highest concentration tested (23.6 μM; Table 1) so an $IC_{50}$ could not be determined but the trend was towards a 1,000-fold lower potency than for DARPin K27 inhibition of the Ras biochemical coupled assay. Moreover, DARPin K27 potently inhibited the MANT-GDP nucleotide exchange assay, whereas effects of DARPin K55 were weak (Supplementary Fig. 4).

In summary, the biochemical characteristics of DARPin K55 are similar to those of the single domain antibody iDab6 (ref. 15) with a strong preference for active, GTP-bound Ras and an ability to directly compete at the Ras/Raf interface. DARPin K27 however shows novel properties for a non-covalent protein inhibitor of Ras, since it binds preferentially to inactive Ras and acts by nucleotide exchange inhibition, thus providing a new tool for investigating this mechanism of action.

**DARPin activity on Ras isoforms and mutants.** In addition to K-Ras wild type, the binding kinetics were measured for DARPin K27 on different Ras isoforms and mutants, with essentially identical dissociation constants obtained on K-Ras G12C, G12D and G12V forms and the N-Ras wild type and G12D forms (Supplementary Table 1). DARPins K27 and K55 also showed consistent potency values when assayed in the Ras biochemical coupled assay or Ras/Raf assay with the K-Ras wild type, G12C, G12D and G12V forms and the N-Ras wild type, G12D, Q61K and Q61R forms (Supplementary Table 2). These data suggest that both DARPins exert their inhibitory effects regardless of the Ras isoform or mutant status.

**Structures of K27 and K55 DARPins binding Ras.** DARPins K27 and K55 were crystallized in complex with the form of K-Ras

G12V to which they bound preferentially, the GDP-bound form for DARPin K27 and the GTPγS-bound form for DARPin K55 and the X-ray crystal structures determined (see Table 2 for data collection and refinement statistics).

DARPin K27 binds to the inactive GDP-bound conformation of K-Ras G12V burying most of switch 1 and leaving most of switch 2 exposed in the inactive conformation (Fig. 2). The binding site of DARPin K27 overlaps that of the GEF, Sos. However, in contrast to the Ras/Sos complex[11], the binding of K27 leaves both the switch 1 loop and the bound GDP in near-identical conformational states to the non-liganded, GDP-bound Ras, whereas Sos binding distorts switch 1, breaks the network of nucleotide contacts and favours nucleotide dissociation. An overlay of the K27/Ras G12V complex with the non-liganded structures of Ras G12V in the active conformation suggests that the active, GTP-bound state is not readily compatible with K27 binding. In particular, the phenylalanine in DARPin K27 at position 56 would appear to clash with the main chain of active Ras in the switch 1 region (Fig. 2b; inset), whereas the switch 1 regions of non-liganded inactive Ras and K27 bound inactive Ras show near identical main chain conformations. Although it may be possible, through an induced fit, for active Ras to accommodate K27, the structural data support a strong preference for K27 binding the inactive form, in agreement with the biochemical data, which would enable it to sequester Ras within the cell in its inactive conformation.

DARPin K55 interacts with both switch 1 and 2 loops of GTP-bound K-Ras G12V in a tight binding interface including ionic interactions at switch 1 (Fig. 3). DARPin K55 does not bind the inactive conformation, in which the switch loops are in different orientations. Comparison of the structure of K-Ras bound to Raf or K55 reveals similar binding interactions with the orientation of the switch loops essentially identical. DARPin K55, therefore, like scFv6 and iDab6 (ref. 15) binds to switch 1 in a manner resembling Raf. Indeed, there are comparable ionic, hydrophobic and hydrogen bond interactions at the interfaces of scFv6, Raf and K55 with switch 1 of Ras (Supplementary Table 3). This strengthens the value of DARPin K55 as a comparator for the behaviour of DARPin K27, which has a different mode of action as a nucleotide exchange inhibitor.

Comparison of the direct residue interactions in the two DARPin/Ras complexes (Fig. 4) highlights that only a minority of Ras residues participate in both DARPin interactions, namely Gln 25, Asp 38 and Ser 39. Most of the remaining Ras residues which contribute to the interface are unique to either the K55 or K27 complex, suggesting in each case a very different overall binding interaction, as supported by the biochemical data.

The crystal structure of DARPin K27 in complex with Ras was used to design a non-binding variant of DARPin K27, lacking the ability to bind Ras, by alanine replacement of three interface residues to make K27 Null3 (Supplementary Fig. 2). K27 Null3 showed negligible binding in the K-Ras biochemical coupled assay (Supplementary Fig. 3).

**Engagement of Ras target within the cell.** The effects of DARPin K27 on Ras within the cell were investigated initially by transient expression from plasmids in comparison with K55 and the control DARPin E3_5. Subsequently, HCT116 cells were stably transduced with FLAG-tagged DARPin K27 on a lentiviral vector and effects compared with and without induction of expression by doxycycline.

A mammalian 2-hybrid system[15] was used to test that DARPin K27 was able to engage with its target Ras intracellularly, since some antibodies are unstable in the reducing environment of the cytoplasm[26]. DARPin K27 was transiently expressed

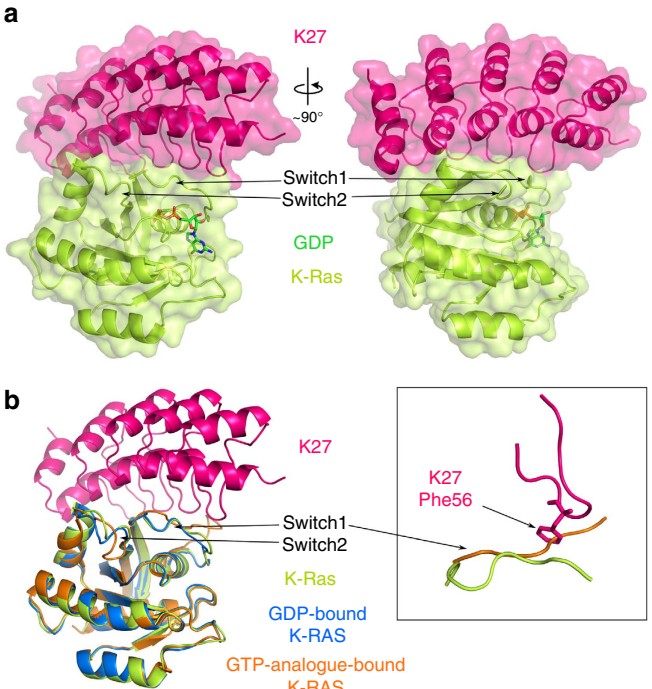

**Figure 2 | Crystal structure of the nucleotide exchange inhibitor DARPin K27 binding to K-Ras G12V at 3.2 Å resolution.** DARPin K27 is shown binding to the inactive form of K-Ras G12V, crystallized in the presence of GDP (**a**). Two views of the complex are shown, rotated at 90° about a vertical axis, with switch 1 and 2 regions annotated. Comparison of the Ras G12V/DARPin K27 complex with unbound Ras G12V structures in the GDP (PDB code: 4TQ9) and GTP analogue-bound (PDB code: 4EFM) states (**b**). Ras in the DARPin K27 complex (green) aligns well with unbound GDP Ras (blue) particularly in the switch 1 loop, whereas GTP analogue-bound Ras (orange) shows the expected conformational change in the switch 1 loop which predicts a clash with the phenylalanine sidechain at position 56 in K27 (see inset).

intracellularly as a fusion to the VP16 transcriptional activator protein while H-Ras, which is highly homologous to K-Ras in the regions bound by the DARPins, was fused to the Gal4 DNA binding domain (Fig. 5a). DARPin K27 interacted with H-Ras as determined by the triggering of luciferase expression, promoted by the interaction of VP16 with the DNA binding domain, compared to the irrelevant control DARPin E3_5 (Fig. 5b). DARPin K55 and other DARPins inhibiting nucleotide exchange or the Ras–Raf interaction also significantly activated luciferase. Engagement of DARPin K27 with native, endogenous K-Ras was then demonstrated in a pull down experiment in HCT116 cells (heterozygous for K-Ras G13D), which had been stably transduced with FLAG-tagged DARPin K27 on a lentiviral vector followed by induction of expression with doxycycline. Following pull down from a cell lysate with anti-FLAG antibody and elution of the tagged DARPins with a FLAG peptide, Ras protein could be detected using K-Ras and pan-Ras specific antibodies, indicating that Ras had been bound to FLAG-tagged DARPin K27 (Fig. 5c). In summary, DARPin K27 can bind to endogenous Ras in its native form within the cell.

**Inhibition of Ras interactions within the cell.** To understand the effects of DARPin K27 on active Ras within cells, an assay was performed to detect inhibition of bioluminescent resonance energy transfer (BRET) between full length K-Ras G12D fused to a modified *Renilla* luciferase (RLuc8) and Raf1 Ras-binding

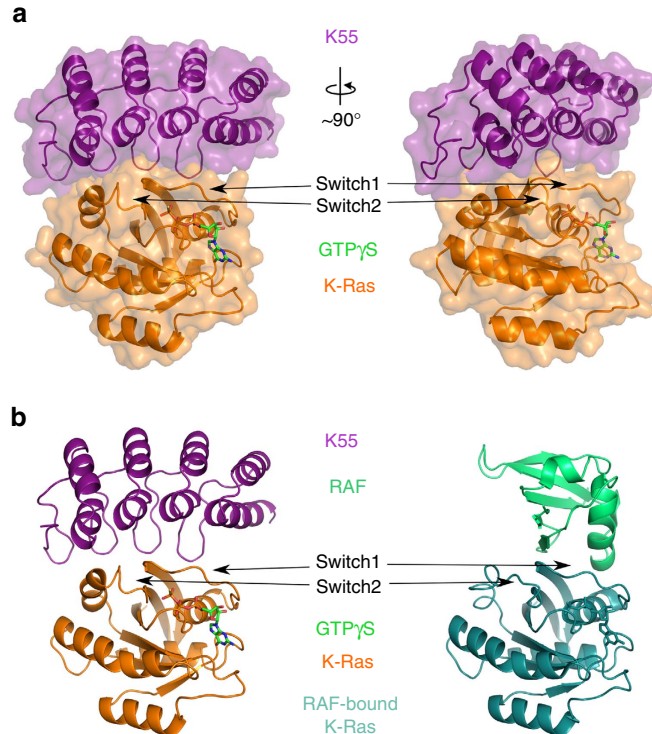

**Figure 3 | Crystal structure of the Ras/Raf inhibitor DARPin K55 binding to K-Ras G12V at 2.3 Å resolution.** DARPin K55 is shown binding to the active form of K-Ras G12V, crystallized in the presence of an uncleavable GTP analogue to ensure an active conformation. Two views of the complex are shown, rotated at 90° about a vertical axis, with switch 1 and 2 regions annotated (**a**). A comparison of the K-Ras/K55 complex with the KRas/RAF complex (PDB code: 3KUD) in **b** shows that the RAF and K55 binding sites overlap significantly and the switch1 loops of K-Ras in each complex are in essentially identical conformations.

domain (RBD) fused to a modified green fluorescent protein (GFP2; Fig. 6a). In parallel, inhibition was performed with two additional RBDs as BRET partners for K-Ras G12D, namely PI3K-α and PI3K-γ and the Ras-associated domain RalGDS. Intracellular expression of DARPin K27 profoundly inhibited BRET between K-Ras G12D and all four downstream effectors, namely PI3K-α, PI3K-γ, RalGDS and Raf1, compared to the negative control DARPin E3_5 (Fig. 6b). Inhibition by DARPin K27 was similar to, or greater than, the inhibition by Ras/Raf inhibitor DARPin K55, also expressed cytoplasmically. Since DARPin K27 acts by nucleotide exchange inhibition the reduction in the BRET signal by this scaffold protein is most likely to be due to the conversion of the cellular K-Ras G12D mainly into the inactive GDP form. As the GTP form is slowly hydrolysed to GDP, the DARPin K27 would inhibit the conversion back into the GTP form by blocking Sos-catalysed nucleotide exchange. Since DARPin K55 is a more potent inhibitor of Ras/Raf than DARPin K27, this supports the effects of DARPin 27 on the BRET signal, being primarily mediated by nucleotide exchange inhibition. In the light of similar studies with single domain antibody iDab6, which showed a dependence on membrane localization for activity[16], it is of interest to note that the DARPins were strong inhibitors even when expressed without membrane localization motifs.

**Inhibition of signalling downstream of Ras.** The ability of DARPins K55 and K27 to inhibit downstream signalling from endogenous Ras was investigated initially by dual transfection of

HEK293 cells with plasmids encoding K-Ras and the relevant DARPins. Western blot analysis confirmed that both K27 and K55, but not control DARPins, were able to reduce levels of ERK and AKT phosphorylation (Fig. 7a) 30 h after transfection. While we did observe a convincing reduction in pERK and pAKT signal for K27 and K55, we also saw variable levels of DARPin expression (Fig. 7a), possibly because the transfection was not equally efficient in all cells in the population. Therefore we switched to a flow cytometry-based assay for subsequent analysis of HCT116 colorectal cancer cells, in order to gate on cells which had been successfully transfected to express a FLAG-tagged DARPin. In the HCT116 cells, active Ras was generated by engagement of the EGF receptor for signalling through wild-type Ras or was constitutively activated for signalling through the mutant allele, leading to a signalling cascade in which ERK phosphorylation could also be detected by flow cytometry. Following transient transfection of a plasmid encoding FLAG-tagged DARPin K27, and gating on FLAG-positive cells, pERK was detected at much lower levels in the absence of EGF stimulation, compared to a control transfection with DARPin E3_5 in the same vector. The response of pERK to EGF stimulation was also strongly inhibited by transient K27 expression (Fig. 7b). Note that when the FACS distribution is examined, the population of cells with high levels of pERK is dramatically reduced. Transfection with plasmid encoding DARPin K55 led to similar reductions in pERK to K27 without EGF stimulation, and again abrogated the pERK response compared to DNA encoding the E3_5 control. To further investigate the pERK response, lentiviral DNA encoding DARPin K27 was stably integrated into the genome of HCT116 cells and expression induced with doxycycline. Levels of pERK, in the absence of EGF, were reduced as compared with cells not induced with doxycycline and again the response of pERK levels to EGF stimulation was reduced (Fig. 7c). These experiments are consistent with inhibition of nucleotide exchange by DARPin K27, leading to accumulation of Ras in the inactive GDP form within the cell and reducing signalling through phosphorylation of ERK.

**Inhibition of Ras-dependent cell proliferation.** The ability of DARPins K27 and K55 to reduce anchorage-independent growth of cells was tested using doxycycline-inducible intracellular expression of the DARPins in a HCT116 cell background. Expression of DARPins K27 and K55 upon doxycycline induction resulted in significant decreases in the ability of the cells to grow in soft agar colony formation assay (Fig. 8). K27 demonstrated a greater reduction in anchorage-independent growth than K55. The HCT116 parental cell line did not show significant decreases in cell colony formation or growth in the presence of the doxycycline treatment.

**Discussion**
The antibody mimetic protein, DARPin K27 has been generated by phage display to provide an inhibitor which non-covalently blocks nucleotide exchange by Ras, showing consistent effects across different isoforms of Ras, including wild type and various oncogenic mutant forms. DARPin K27 binds preferentially to the inactive GDP form, with an epitope close to the switch regions in their inactive conformation. Although DARPin K27 binds weakly to the K-Ras GTPγS form, inhibition of Ras/Raf binding is approximately three orders of magnitude weaker than inhibition of a coupled assay linking nucleotide exchange to Ras/Raf binding. DARPin K27 provides a new tool for understanding the effect of nucleotide exchange inhibitors, in particular the critical step of GTP loading, on Ras function within the cell. The comparator DARPin K55 has also been isolated, which interacts

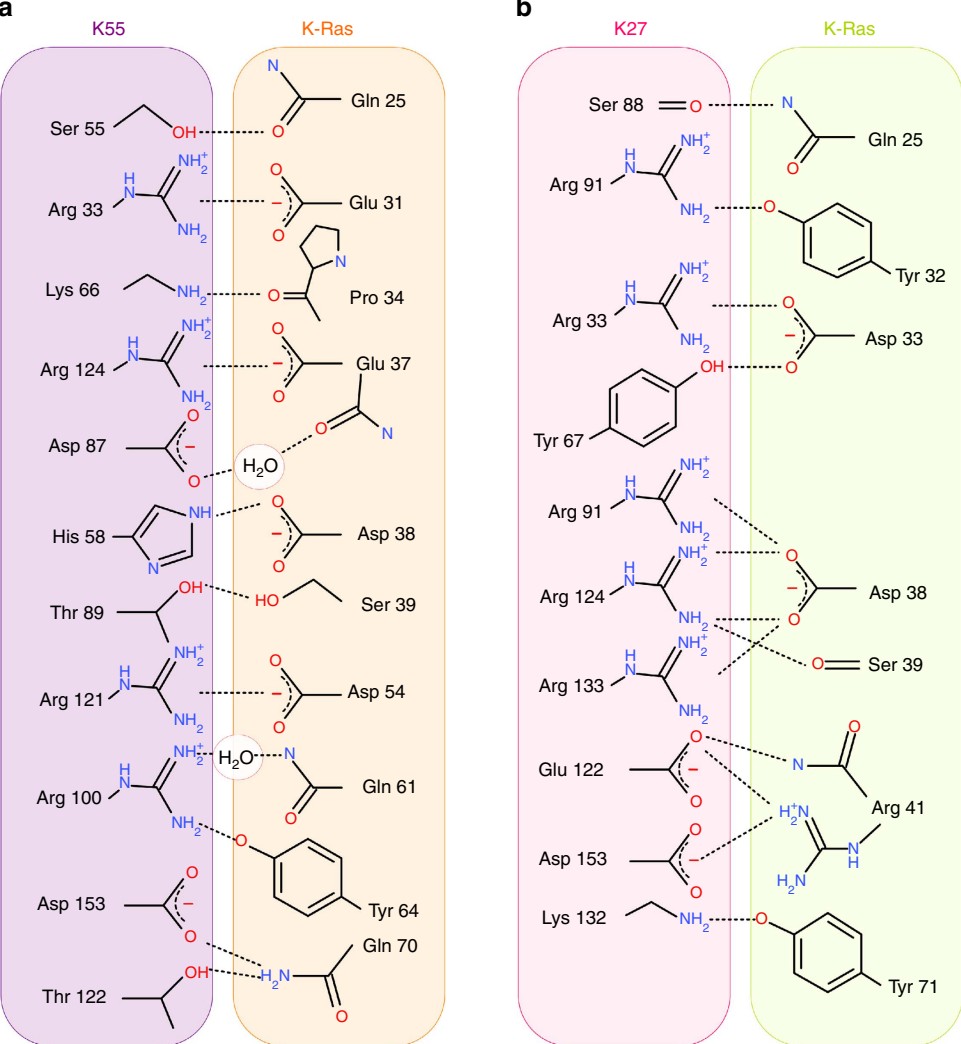

**Figure 4 | Comparative interaction summaries for K27 and K55 complexes with Ras.** Direct residue interactions in the two DARPin/Ras complexes are summarized for the comparison of DARPin K55/Ras (**a**) with K27/Ras (**b**). While some Ras residues participate in both DARPin interactions, namely Gln 25, Asp 38 and Ser 39, most of the participating Ras residues are unique to either the K55 or K27 interaction, suggesting very different overall binding modes in each case.

with active Ras at switch 1 in a very similar way to Raf and the previously described scFv6 and iDab6.

Earlier studies have reported that non-covalent nucleotide exchange inhibitors can reduce EGF stimulated pERK signalling through wild-type Ras. Patgiri et al.[19] reported that a constrained helix, HBS3, derived from Sos (929–944) bound to nucleotide-free Ras with a $K_d$ of 28 μM and to GDP-bound Ras with a $K_d$ of 158 μM. Perhaps surprisingly, given this very weak binding, HBS3, at 75 μM in the extracellular media, was reported to reduce pERK signalling following EGF stimulation of HeLa cells. Leshchiner et al.[20] prepared a stapled helix SAH-SOS1$_A$, also derived from Sos (929–944). SAH-SOS1$_A$ bound to Ras with an EC$_{50}$ in the range of 100–175 nM and inhibited MANT-nucleotide exchange. Leshchiner[16] reported SAH-SOS1$_A$ at concentrations over 10 μM reduced phospho-signalling following EGF stimulation of Panc10.05 and HeLa cells. DARPin K27 is a much more potent non-covalent nucleotide exchange inhibitor than HBS3 and SAH-SOS1$_A$. Moreover, DARPin K27 has been shown to engage Ras within the cell, giving the opportunity, following expression intracellularly, to more completely shut down native nucleotide exchange and investigate the consequent impact on reducing the amounts of

EGF stimulated active wild-type Ras and constitutively active mutant Ras.

We have demonstrated that DARPin K27 can engage with native K-Ras within the cell and that it can inhibit the K-Ras interaction with Raf and consequent downstream signalling leading to phosphorylation of ERK. Effects on the BRET signal from K-Ras G12D binding to Raf have been observed and marked effects are seen on pERK levels in cells that are heterozygous for Ras. In a recent review, Stephen et al.[2] speculated that oncogenic Ras exists in a nucleotide free state frequently enough to make it vulnerable to attack. Recent work from Patricelli et al.[21] and Lito et al.[22] addressed this question using a covalent, small molecule inhibitor, ARS-853, of G12C mutant Ras that does not inhibit wild-type Ras. These authors showed that ARS-853 can induce dramatic reductions in active Ras G12C levels and downstream signalling, despite targeting only GDP-bound KRas G12C. Their data support K-Ras G12C not being locked in a fully activated state, but, rather, rapidly cycling its nucleotide exchange state, allowing ARS-853 to achieve complete binding and inhibition over time as Ras cycles its bound nucleotide. More generally, Stephen et al.[2] raised the question of whether nucleotide exchange inhibitors would reduce the amount of

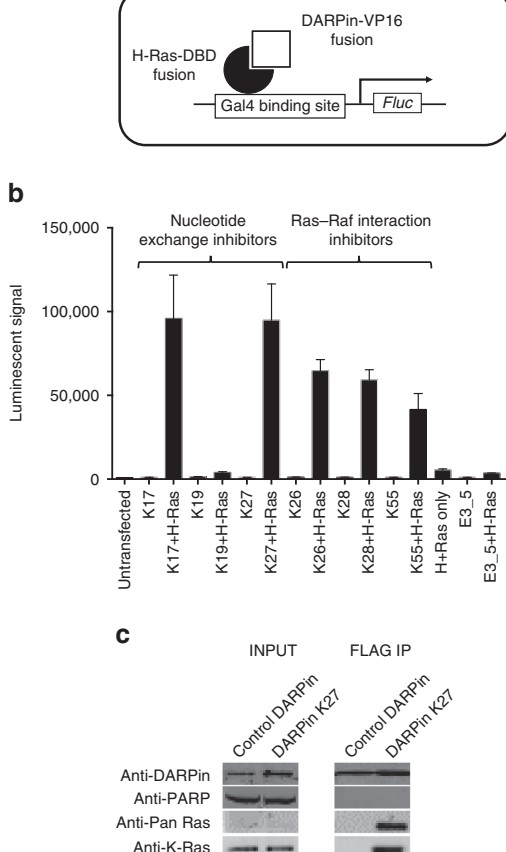

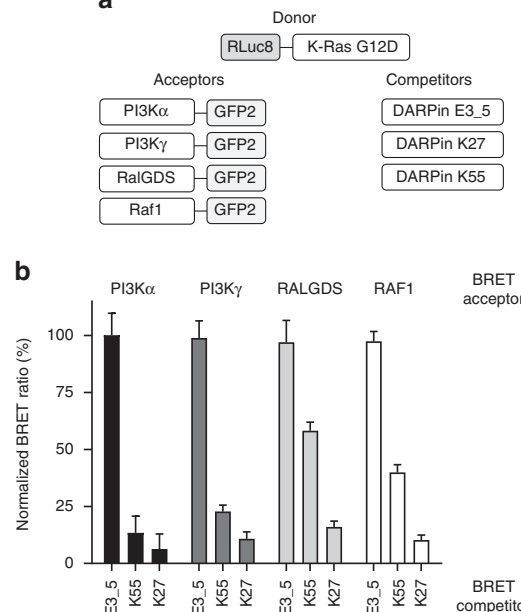

**Figure 6 | Anti-Ras DARPins inhibit intracellular Ras interactions with downstream effector molecules.** In an intracellular bioluminescent resonance energy transfer (BRET) assay, full-length K-Ras G12D-RLuc8 donor molecules were co-expressed with various Ras-binding or Ras-associated domains fused to GFP2 acceptors to trigger a BRET signal. (**a**) Diagrammatic format of donor, acceptor and competitor molecules. (**b**) Normalized BRET ratio (normalized to the negative control DARPin E3_5) generated in HEK293T cells transfected with plasmids expressing the indicated BRET pairs in the presence of different DARPins as competitors. Error bars represent the mean ± s.d. (n = 2).

**Figure 5 | Anti-Ras DARPins engage Ras within the intracellular environment.** In a mammalian two-hybrid assay, interaction of an H-Ras-DNA-binding domain fusion and a DARPin-VP16 transactivator fusion, following transient transfection of expression plasmids, enables binding to a Gal4 binding site and activation of a firefly luciferase reporter in the cell line CHO-Luc15 (**a**). Luciferase signals resulting from co-expression of various DARPin and H-Ras fusion pairs (**b**) demonstrate that the majority of anti-Ras DARPins interact with H-Ras within the reporter cell, regardless of their inhibitory mechanism or preference for inactive or active Ras. Error bars represent the mean ± s.d. (n = 3). FLAG-tagged DARPin expression in an inducible HCT116 cell line followed by FLAG-tag immunoprecipitation demonstrates that DARPin K27, but not a control DARPin, can bind and capture endogenous Ras within a tumour cell line (**c**).

active wild-type Ras in preference to active mutant Ras and suggested that this required investigation with sustained exposure within mutant cells to nucleotide exchange inhibitors. The data in our study suggest that non-covalently binding nucleotide exchange inhibitors can reduce the amount of active mutant K-Ras, as seen in the BRET assay for G12D. Moreover, in the pERK assay, the amount of active Ras in HCT116 cells, without EGF stimulation, is reduced by K27, possibly reflecting effects on Ras G13D, and also the response to EGF is reduced, probably reflecting mainly effects on wild-type Ras. Note, however, that Patricelli et al.[21] reported that EGF stimulation can support active KRas G12C levels, so some of the EGF response may reflect signalling through G13D. The pERK response without addition of EGF in our experiments may in part be due to serum components in the media, but the effects of DARPin K27 on the signal in the BRET assay for Ras/Raf interaction suggest that DARPin K27 would also be suppressing constitutive pERK signalling via Ras G13D. Since HCT116 cells are dependent on active Ras for

growth and proliferation, reduction in active Ras may be expected to promote apoptosis. A decrease in growth is observed in the soft agar colony formation assay on expression of DARPin K27, in accord with the profound K-Ras dependence noted by Patricelli et al.[21] in anchorage independent settings.

The effects of intracellular DARPin K27 on BRET signal and the pERK assay parallel those of DARPin K55, which directly inhibits the Ras/Raf interaction. DARPin K55 is a close mimic of iDab6, which has been shown to reduce tumour growth on induction of its expression in tumours formed from HCT116 cells. Since the two DARPins are structurally and biophysically similar, this pair of molecules can act as useful tools to systematically explore the two mechanisms of Ras inhibition in different cancer cell backgrounds.

To understand the potential for inhibition of nucleotide exchange as a therapeutic approach to Ras-dependent tumours, there are a number of further studies that would be valuable using intracellular expression of DARPin K27. It would be desirable to more directly test the effects of nucleotide exchange by K27 on different mutant forms of K-Ras, each with different intrinsic hydrolysis rates, by studying cells containing mutant but not wild-type Ras alleles, perhaps prepared by CRISPR techniques. Further, it would be valuable to test the effect of DARPin K27 inhibition of nucleotide exchange on growth in vivo of tumours dependent on Ras, such as HCT116 cell xenografts, following doxycycline induced expression within the HCT116 cells.

There remain formidable barriers for therapeutic intervention by non-covalent inhibition of Ras nucleotide exchange. The first arises from the intracellular concentration of Ras, the total of which has been estimated as 0.4 μM for HeLa cells and 0.53 μM for COS-7 cells[27]. This is consistent with the determination that

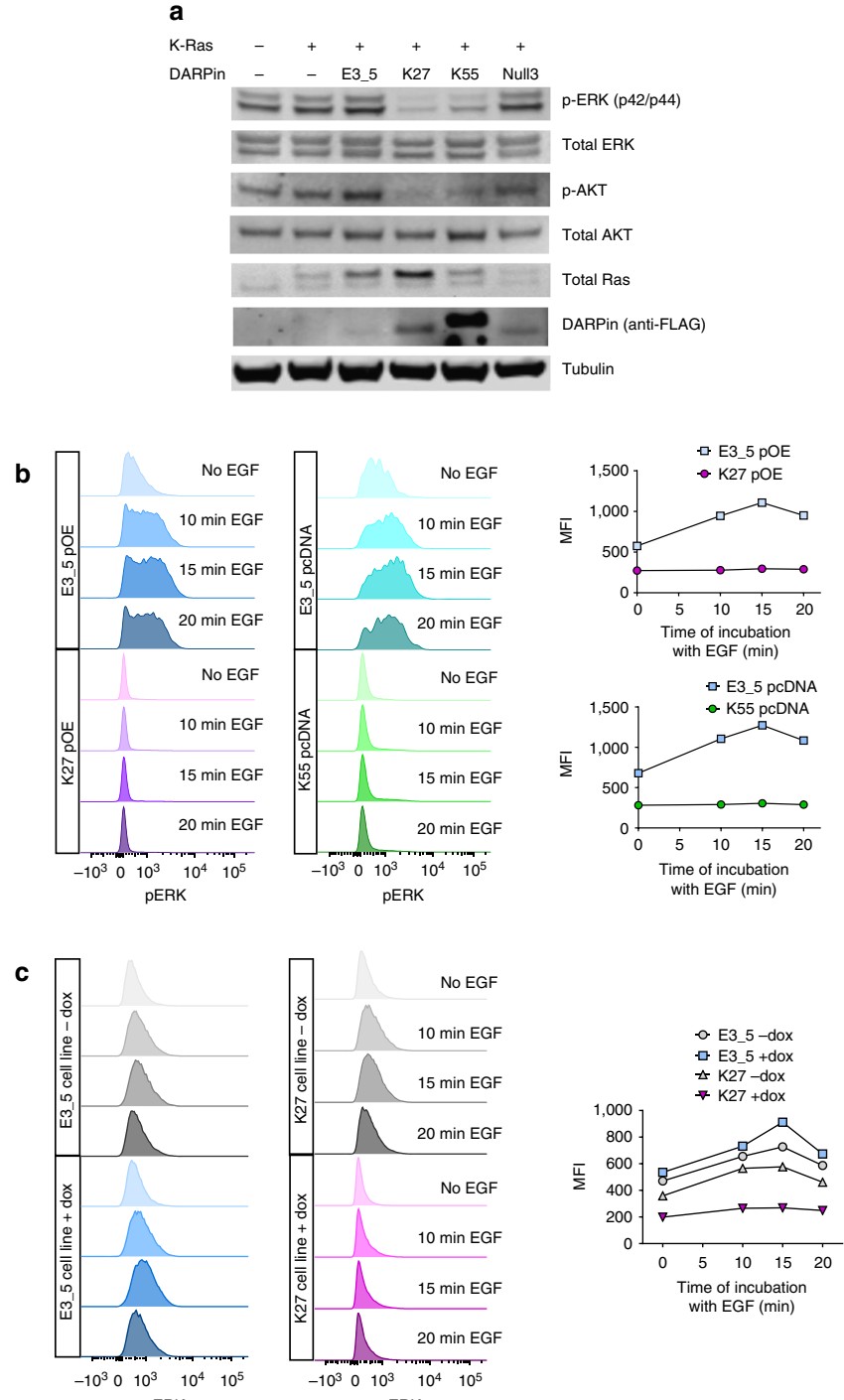

**Figure 7 | Anti-Ras DARPin K27 and K55 expression in HEK293 and HCT116 cells causes inhibition of ERK phosphorylation.** HEK293 cells were co-transfected with plasmids encoding (i) K-Ras and (ii) DARPins to study changes in ERK phosphorylation (**a**). Thirty hours post-transfection, HEK293 cells were lysed for Western blot analysis with the antibodies indicated. For flow cytometric measurement of ERK phosphorylation, HCT116 colorectal cancer cells expressing K27, K55 or control DARPin E3_5 were stimulated with EGF for 0, 10, 15 or 20 min before analysis. For each condition histograms of phosphorylated ERK (pERK) are shown and median fluorescence intensity (MFI) is plotted. HCT116 cells were collected either 24 h post transfection with plasmids encoding the DARPins, and subsequently gated on cells expressing high levels of FLAG-tagged DARPins (**b**) or HCT116 cell lines with lentivirally transduced DARPin gene under a doxycycline-inducible promoter were used and assayed after 48 h of culture with or without doxycycline (**c**).

SW48 colorectal cancer cells contain greater than 260,000 total Ras protein copies per cell, with a subset of oncogenic K-Ras mutants exhibiting increased total cellular Ras abundance and an increased ratio of mutant versus wild-type K-Ras protein[28]. Intracellular concentrations of nucleotide exchange inhibitor in excess of the Ras concentration may be required to largely suppress the amount of active Ras GTP form. In practice, Ras compartmentalization, for example, into membrane clusters[29] may mean that the total Ras concentration is not reflective of the amount that needs to be neutralized. Further, Ras mutants in which the intrinsic rate of GTPase activity is better retained, such as G12C and G12D (ref. 30) are likely to be more rapidly

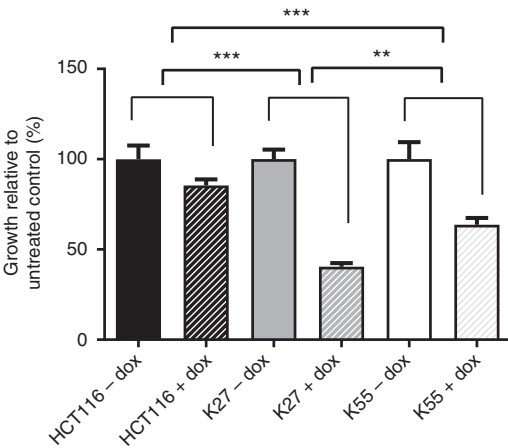

**Figure 8 | Effect of intracellular expression of K27 and K55 on anchorage-independent growth.** Parental HCT116 cells and lentivirally transduced inducible K27 and K55 HCT116 cells were grown in a soft agar colony formation assay in the presence and absence of $1\,\mu g\,ml^{-1}$ doxycycline for 14 days. Anchorage-independent growth was measured by alamar blue assay fluorescence and for each cell line was normalized to its respective untreated control ($n = 4$). The statistical significance of the difference in growth signal observed in response to the addition of doxycycline was calculated by ANOVA followed by $t$-test adjusted for multiple comparisons using Bonferroni correction where $**P < 0.01$ and $***P < 0.001$.

inactivated by nucleotide exchange inhibitors than Ras mutants with a lower intrinsic rate such as G12A and Q61H.

A second barrier arises from the fact that Ras is intracellular. DARPin K27 has no intrinsic ability to enter cells and therefore cannot access Ras when the DARPin is added extracellularly. Although there have been reports of delivery of DARPins to the cytoplasm of the cell, facilitated by the Pseudomonas exotoxin A translocation domain[31] or by anthrax toxin elements[32], substantial increases in efficiency would be required to make the approach viable therapeutically. It may be challenging to develop a small molecule inhibitor binding at the same site as DARPin K27, since the scaffold binds across a broad surface, rather than defining a pocket. However, there are potentially other approaches to delivery of genes encoding protein-based Ras inhibitors, including the rapidly developing fields of oncolytic virus[33] and mRNA delivery[34].

In conclusion, the data reported here support nucleotide exchange inhibition as a valid mechanism of action to block Ras signalling in tumour cells and expand on previous studies which were restricted to G12C Ras mutants. We have extensively characterized DARPin K27, a valuable tool for understanding nucleotide exchange inhibition of multiple Ras isoforms and mutants in different cellular contexts, along with DARPin K55, a positive control for Ras/Raf inhibition, and provide a negative control DARPin K27 Null3. Protein-based tools such as these DARPins, and the recently discovered allosteric Ras inhibitor NS1 (ref. 35), will be important in fully understanding the potential of novel Ras inhibition strategies and will hopefully move us closer to developing Ras-targeted therapies in the clinic.

## Methods

**Ras protein.** The human *KRAS* gene sequence (residues 1–166, Isoform 2B, P01116-2) containing the G12V substitution was synthesized by Geneart (Life Technologies) and cloned into a pET28b vector with an N-terminal His6 tag. Sequences of *KRAS*, *NRAS* and *HRAS* DNA constructs expressed in this study are shown in Supplementary Table 4. DNA constructs encoding other Ras isoforms and mutants were synthesized and cloned in similar fashion. Ras proteins were expressed in BL21 (DE3) *Escherichia coli* (New England Biolabs) and cell pellets

resuspended in 50 mM Hepes (pH 7.4), 100 mM NaCl, 20 mM Imidazole, 2 mM TCEP and 2 mM $MgCl_2$. A tablet of cOmplete EDTA-free protease inhibitor was also added (Roche—cat #11873580001) and following sonication the sample was centrifuged at 20,000 r.p.m. and proteins purified from the supernatant by Ni-NTA chromatography on a 5 ml HisTrap HP column (GE Life Sciences—cat # 17-5248-02). Following elution in 50 mM Hepes (pH 7.4), 500 mM NaCl, 400 mM Imidazole, 2 mM TCEP and 2 mM $MgCl_2$, samples were further purified by size exclusion chromatography on a Superdex 75 column. Final buffer composition was 50 mM HEPES (pH 7.4), 100 mM NaCl, 2 mM $MgSO_4$. K-Ras and N-Ras (1–172) G12V, G12C, G12D, Q61K, Q61R and wild-type proteins were biotinylated via an Avi tag using BirA enzyme and then exchanged with either GDP or GTPγS, catalysed by Sos (Son of sevenless). The GTPγS in the K-Ras GTPγS had a tendency to be hydrolysed to GDP on storage, so the levels were determined by electrospray mass spectroscopy.

**DARPin selection and purification.** DARPins were isolated from a phage display library containing $1 \times 10^9$ members[36]. The first two rounds of selection were performed by a panning method in which 250 nM biotinylated K-Ras GTPγS (1–166) was pre-bound to streptavidin magnetic beads. The phage library was applied, beads were washed to remove unbound phage and bound phage eluted with trypsin. For round 3, a soluble selection method was used in which the phage was bound to 25 nM K-Ras GTPγS and bound phage captured using streptavidin magnetic beads, before washing and elution of bound phage with trypsin. For characterization, DARPins were subcloned to the pET16b vector, using the oligonucleotides pET16b_DP_Fwd (5′-CGATCATATGGATCTGGGAAAAAAAC TGCTGGA-3′) and pET16b_DP_Rev (5′-ATCGGGATCCTCACTACAGTTTCT GCAGG-3′) and expressed cytoplasmically in BL21 (DE3) *E. coli* (New England Biolabs). Following lysis in BugBuster plus Benzonase (EMD Millipore), DARPins were purified to homogeneity using nickel-chelate chromatography, followed by size exclusion chromatography to provide a monomeric protein in PBS (pH 6.5). DNA sequences of DARPins expressed in this study are shown in Supplementary Table 4. An HA-tag was fused to the C-terminus of DARPin K55 to provide residues with an absorbance at 280 nm, since the original DARPin lacked Tryptophan and Tyrosine residues.

**Immunoassays for Ras binding.** DARPins from selections were tested for their ability to bind K-Ras GTPγS or K-Ras GDP by ELISA with phage displaying DARPins or DELFIA (dissociation-enhanced lanthanide fluorescence immunoassay) using DARPins solubly secreted into the periplasm using the methods described in Vaughan et al.[37]. Specificity was assessed by assay for lack of significant binding to streptavidin.

**Ras biochemical coupled assay.** The Ras biochemical coupled assay was an adaptation of the HTRF assay used by Winter et al.[38]. A coupled reaction was performed linking nucleotide exchange catalysed by Sos, to the Ras/Raf interaction detected by HTRF. Biotinylated Ras was pre-incubated with streptavidin-Europium chelate to form a complex, and in a separate reaction, Raf-GST (glutathione-S-transferase) was pre-incubated with anti-GST-XL665 to form a second complex. Dilution series of samples (in PBS containing 0.1 mg ml$^{-1}$ BSA) and controls were loaded into an assay plate and the streptavidin:Ras complex added, followed by incubation for 15 min. Nucleotide exchange was initiated with the addition of GTPγS and SOS and the Raf:anti-GST complex was added. Final concentrations added were 2 nM biotinylated Ras; 37.5 ng ml$^{-1}$ streptavidin-Europium; 12 nM Raf-GST; 2 µg ml$^{-1}$ Anti-Raf GST XL665; 4 µM GTPγS and 2 µM Sos. In addition the buffer for the Raf:anti-GST complex contained 0.1 mg ml$^{-1}$ BSA and 0.1 M potassium fluoride. After 1 h incubation, FRET (Fluorescence Resonance Energy Transfer) was measured on an Envision plate reader at emission wavelengths 620 and 665 nm. Ratiometric reading was calculated by dividing counts at 665 nm by counts at 620 nm, then multiplying by 10,000. Responses were plotted in Prism. Curves were fitted using non-linear regression, equation: log (inhibitor) versus response, variable slope.

**Ras/Raf assay.** An HTRF assay for the interaction of Ras/Raf was similar to the Ras biochemical coupled assay above, with the exceptions that biotinylated Ras GTPγS was pre-incubated with streptavidin-Europium-cryptate and buffer was added rather than GTPγS and SOS.

**MANT assay.** Exchange of GDP with dMANT-GDP (2′-Deoxy-3′-O-(N′-methylanthraniloyl)guanosine-5′-O-diphosphate) was studied in an assay adapted from that described in Winter et al.[38]. DARPins were tested for inhibition of nucleotide exchange by incubating at 10 µM with 500 nM K-Ras, 1 µM SOS and 500 nM dMANT-GDP (Biolog, #D084-05). All reagents were diluted in 20 mM HEPES pH 7.4, 150 mM NaCl, 5 mM $MgCl_2$, 0.1% Octyl glucoside, 1 mM DTT. Exchange reactions were performed in 384-well low volume black plates (Greiner #784076). The increase in fluorescence of dMANT-GDP upon binding to K-Ras as a result of SOS mediated nucleotide exchange was detected using an Envision plate reader (Perkin Elmer) using 340 nm/450 nm excitation/emission filters. Data were captured at time zero (T0) and after 60 min (T60). The 60 min

data were normalized to the time zero data (T60–T0) to account for well-to-well variability and background fluorescence subtracted (KRas negative controls). % inhibition was calculated for the DARPin treated samples: [(buffer control response − DARPin treated response)/(buffer control response)] × 100.

**DARPin kinetic binding measurements.** ForteBio Octet Red384 was used to determine the binding kinetics initially using seven concentrations of the DARPins from 3,000 nM to 4.1 nM in 1/3 dilution steps in buffer (PBS containing 1 mg ml$^{-1}$ BSA and 0.01% Tween), with varying concentrations for repeat determinations. For each sensor the following protocol was carried out: Baseline wash in octet buffer for 60 s, loading of biotinylated Ras (1–166; mutant or wild type; GDP or GTPγS) at 5 μg ml$^{-1}$ for 180 s followed by a further 60 s baseline wash. DARPins were then associated for 180 s and dissociated for 600 s by washing in buffer. For each sample a reference well was included only containing buffer. Data were analysed using ForteBio data analysis. Reference wells were subtracted from sample wells and 1:1 global fitting model was used to fit curves to sensorgrams and determine $K_{on}$, $K_{off}$ and $K_d$ values. For measurements of kinetic constants for Ras GDP forms, 2 mM MgSO$_4$ and 100 μM GDP were included in the buffers. For measurements of kinetic constants for Ras GTPγS forms, 2 mM MgSO$_4$ and 100 μM GTPγS were included in the buffers.

**Cell lines.** The cell lines used in this work are not listed in the database of commonly misidentified cell lines, were obtained from reputable sources (ATCC, ECACC) and were tested and found to be mycoplasma-free before use.

**Bioluminescence resonance energy transfer assay.** HEK293T seeded in six-well plates were transfected with 50 ng of donor plasmid (RLuc8-KRASG12D-CAAX) and 100–150 ng of acceptor plasmid (RAF1$_{RBD}$-GFP2, PI3Kα$_{RBD}$-GFP2, PI3Kγ$_{RBD}$-GFP2, RALGDS$_{RA}$-GFP2) and with 100 ng of appropriate competitor plasmid (pEF-DARPin-myc). Cells were detached 24 h later and seeded in a white 96-well plate (clear bottom, PerkinElmer, cat#6005181). Cells were left for an additional 24 h at 37 °C. The BRET2 signal was read directly after addition of Coelenterazine 400a substrate (10 μM final) on cells (Cayman Chemicals, cat#16157) on an Envision instrument (2103 Multilabel Reader, PerkinElmer) with the BRET2 Dual Emission optical module (515 ± 30 nm and 410 ± 80 nm; PerkinElmer). The experiment shown in Fig. 6 was performed with technical triplicates and also repeated once with similar results.

**Structural characterization.** K-Ras (1–166) G12V and Darpin 27 or Darpin55 samples were mixed in equimolar amounts and incubated overnight at 4 °C to allow for complex formation. The complex sample was then concentrated to 16 mg ml$^{-1}$ immediately before crystallization experiments.

K-Ras/Darpin complexes were crystallized using the sitting-drop vapour diffusion method at 293 K. Equal volumes of protein and reservoir solutions were equilibrated against:

K-Ras (1–166) G12V GDP/Darpin 27 complex: 30% (w/v) PEG-MME 2000, 200 mM ammonium sulfate, 100 mM sodium citrate pH 5.6 K-Ras (1–166) G12V GTPyS/Darpin 55 complex: 18% (w/v) PEG8000, 200 mM calcium acetate, 100 mM sodium cacodylate pH 6.5.

Crystals appeared within 24 h and were collected within 3 days. Crystals were cryoprotected in reservoir solution supplemented with 20% (v/v) ethylene glycol before freezing in liquid nitrogen. Data collection was carried out at Diamond Light Source beamlines i03 (K27, at 0.927 Å wavelength) and i04 (K55, at 0.979 Å) using a Pilatus 6M detector. Data were processed using the programs XDS[39] and aimless[40]. Structures were solved with molecular replacement with Phaser[41]. Molecular models were built in Coot[42] and refined in refmac5 (ref. 43). A stereo view of part of the electron density map for each structure is shown in Supplementary Fig. 6.

**Ras immunoprecipitations.** HCT116 DARPin cell lines were constructed using lentiviral vectors encoding DARPins K27 and K55, each fused to a C-terminal FLAG tag, and under the control of a doxycycline inducible promoter. Irrelevant control DARPin CMYC20 was included as a control. Following 48 h induction of DARPin-FLAG expression, cells were lysed and then incubated overnight at 4 °C with anti-FLAG magnetic beads (Sigma, cat#F3290-4MG). After washing, DARPins were eluted with FLAG peptide (Sigma, cat#F3290-4MG) and Ras pulled down was detected in a western blot with an anti-pan-Ras antibody (Cell Signalling cat#3339) or an anti-K-Ras antibody (LSBio cat#LS-C175665). Additional antibodies used were anti-DARPin (in-house mouse monoclonal IgG) and anti-PARP (Cell Signalling cat#9542). The original scans of all western blots are provided as Supplementary Fig. 5.

**Mammalian 2-hybrid assay.** CHO-Luc15 cells were seeded in 96-well plates at $3 \times 10^4$ cells per well in 100 μl of DMEM (10% FBS) and incubated at 37 °C, 5% CO$_2$ overnight before the cells were transfected with plasmids pM1-HRASG12V(DBD) and pEFVP16-DARPin using lipofectamine 2000. The activity of firefly luciferase was measured using Dual-Luciferase Reporter Assay System

(Promega) 48 h after transfection. The experiment shown in Fig. 5 was performed with technical triplicates and also repeated at least two times with similar results.

**pERK analysis by western blot and flow cytometry.** Adherent HEK293 (Ad293) cells were dual transfected with a wild-type K-Ras encoding plasmid (Origene SC109374) and pcDNA3.1 vectors expressing FLAG-tagged DARPins. Thirty hours post-transfection, to allow for transient expression of proteins from each plasmid, cell lysates were prepared. Antibodies used for western blot analysis were anti-Ras (Cell Signalling Technology #3339S, 1:1,000), anti-pERK p44/42 (Cell Signalling Technology #4377S, 1:1,000), anti-ERK p44/p42 (Cell Signalling Technology #9102S, 1:1,000), anti-pAKT (Cell Signalling Technology #3787S, 1:1,000), anti-AKT (Cell Signalling Technology #9272, 1:1,000), anti-FLAG M2 (Sigma #F1804, 1:1,000), anti-rabbit Ig IRDye 800CW (LI-COR #925-32211, 1:15,000) and anti-mouse Ig IRDye 680RD (LI-COR #926-68070, 1:15,000). For flow cytometry, HCT116 cells were transiently transfected with pcDNA3.1 or pOE vectors expressing DARPins K27 or K55 tagged with FLAG at N-terminus. Alternatively, HCT116 cells containing lentiviral stably integrated DARPin constructs with C-terminal FLAG were grown and expression of DARPins induced with doxycycline. Cells were treated with accutase, washed and re-suspended in PBS. Cell suspensions were then divided into four wells and treated with 50 ng ml$^{-1}$ EGF for 10, 15 and 20 min or left untreated in PBS. Next, cells were fixed with 4% formaldehyde and permeabilized with methanol. Finally, cells were incubated with anti-FLAG conjugated to Alexa488 (Cell Signalling cat#5407, 1:1,000) and anti-pERK conjugated to Alexa647 (Biolegend cat#675504, 1:1,000) followed by flow cytometric analysis performed using a FACSCanto II flow cytometer and BD FACSDiva software (BD Biosciences). For cells transiently transfected with FLAG-tagged DARPins, gating for a high FLAG signal was performed before pERK analysis. Data collected were analysed in FlowJo software. Technical replicates were not performed for flow cytometry studies but the experiments in transiently-transfected and stable cell lines were each repeated at least twice with similar outcomes.

**Soft agar colony formation assay.** Soft agar colony formation assay was performed in black walled 96-well tissue culture plates (Greiner). The base layer comprised of 50 μl of 0.7% low melting temperature agarose (ThermoFisher Scientific) containing DMEM, 10% FBS (v/v) and 3 μg ml$^{-1}$ puromycin (ThermoFisher Scientific). The top agar comprised of 50 μl 0.3% agarose containing DMEM, 10% FBS (v/v) and 3 μg ml$^{-1}$ puromycin (ThermoFisher Scientific). HCT116 parental and inducible K27 and K55 HCT116 cells were embedded at 1,000 cells per well in the top agar as single cells. Fifty microliters media feed was added to each well. Doxycycline (Sigma Aldrich) was added to the feed at a final concentration per well (150 μl) of either 0, 1 or 2 μg ml$^{-1}$. 4, 7 and 10 days after seeding, media feed was replaced with 50 μl feed per well (including dox) as above. fourteen days after seeding, cells were subjected to a proliferation assay by the addition of 20 μl per well Alamar Blue (ThermoFisher Scientific). Plates were incubated at 37 °C for 4 h before reading fluorescence on a plate reader (Excitation at 550 nm and Emission at 600 nm). Alamar blue signals for treated wells were normalized to respective untreated control well data. The statistical significance of the difference in growth signal observed in response to the addition of doxycycline was calculated by ANOVA followed by t-test adjusted for multiple comparisons using Bonferonni correction. The experiment shown in Fig. 8 was performed with biological duplicates and also repeated once with similar results.

**Data availability.** Coordinates and structure factors have been deposited at the Protein Data Bank (PDB) with accession codes 5O2S (K27) and 5O2T (K55). Other data supporting the findings of this study are available from the corresponding author on reasonable request.

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

## Acknowledgements

We gratefully acknowledge the contributions of the following people: Sarah Ross and Kevin Hudson (discussions on Ras biology), Ben Kemp, Anna Czyz and Sonia Raithatha (DARPin expression and purification), Richard Stevens and Shaban Javed (DNA chemistry), Alan Sandercock (Octet analysis), Jon Travers (Ras immunoprecipitations), Bruck Taddese (Computational Structural Biology) and Guglielmo Rosignoli (cell sorting). We thank Diamond Light Source for beamline access and technical support.

## Author contributions

S.G., P.K.-Z., J.D., J.B., J.Z., N.B., R. Marwood, J.T., R.O., P.S., B.M. and R.J. planned and executed experiments, analysed the data and were involved in data discussions. R.J. and R.Minter wrote the paper. C.P., T.R., R.J. and R. Minter planned experiments and were involved in data discussions. All authors critically reviewed and approved the manuscript.

## Additional information

**Competing interests:** S.G., P.K.-Z., J.D., J.B., R. Marwood, J.T., R.O., P.S, B.M., C.P., R.J. and R. Minter were employees of the AstraZeneca Group while performing this work and may have stock/stock options in AstraZeneca. The remaining authors declare no competing financial interests.

