## [Peer review file · Nature Communications]

Reviewers' comments:

Reviewer #1 (Remarks to the Author):

The manuscript by Guillard et al. describes the biochemical, structural and cellular characterization of two designed ankyrin repeat proteins (DARPin) in their interaction with Ras proteins. Ras is a major driver in many oncogenic processes and has been a target for a number of biochemical approaches in order to inhibit downstream signaling of oncogenic Ras mutants but also considered undruggable for various reasons. The authors use phage display approaches followed by a Ras biochemical coupled assay to isolate two DARPins, termed K27 and K55, respectively. While K27 bound to GDP-bound RasG12V to inhibit Sos mediated nucleotide exchange, K55 is reported to act to inhibit the interaction between Ras and its downstream target Raf-kinase. The DARPins seem to act independently of the Ras-Isoform or mutant status as demonstrated by testing inhibition of N- and K-Ras along with several known oncogenic mutations. This is in contrast to previously reported compounds that depended on the presence of the G12C -mutation. Nucleotide exchange activity seems blocked primarily by stabilization of the GDP-bound form, as has been reported for small molecule compounds specifically acting on the G12C mutant Ras. However, as mentioned above, in this work DARPin seem to act on several Ras isoforms and be independent of the mutational status of Ras. Crystal structures of Ras-DARPin complexes show the K27 binding site overlapping with that of guanine nucleotide exchange factor Sos and of K55 with that of Raf-RBDs. Intracellular interaction between Ras and DARPins was probed using a mammalian 2-hybrid system with a luciferase readout. Inhibition of Ras-Raf interaction in cells was not dependent on the presence of membrane localization motifs. To corroborate the notion that Ras-inhibition by DARPins affect Ras/MAPK signaling the authors demonstrate reduction of pERK levels after transfection with DARPin encoding plasmids and EGF stimulation of cells. Finally the authors demonstrate DARPin mediated growth inhibition in soft agar colony formation assays.

In summary the paper presents a comprehensive biochemical, structural and cellular analysis of a novel designed Ras inhibitory protein scaffold. The structures of Ras-DARPin complexes define for the first time the interface between Ras and designed proteins that inhibit nucleotide exchange and act on a relatively broad variety of Ras variants or mutants. The research is important especially in light of a number of other studies, most importantly the small molecule Ras inhibitor rigosertib, after a long time of silence in the Ras-drugging field. Exploration of the molecular details of the interaction may enable design of chemical compounds/components that act by a similar mechanism and understand the mechanistic details of the interaction. The technical quality is high, results and methods are presented clearly, conclusions are justified, discussion appropriate.

I recommend publication of the paper in Nature Communications considering the following comments/suggestions:

P2: L 40ff: Original switch definition according to Milburn et al. (1990) is: switch I, res. 30-38; switch II, 60-76. Regarding approaches to modulate Ras function, increase of GTPase activity may also be an approach for modulating Ras function that should be particularly valid in view of the observations that most oncogenic Ras mutant show impaired GTPase activity. Consider this aspect for introduction/discussion.

P7: L148: Why was H-Ras used in the 2-hybrid assay and not K-Ras as for the in vitro interaction studies?

Fig. 1: rotation axis and angle connecting panels in a) should be included in figure. I do not recognize RasGDP in purple. The DARPin-Ras interface is seems of the most important features of the paper and should therefore be shown in more detail along with a cartoon specifying the interactions (e.g. similar like in Tanaka et al., 2007, ref.) for the biologist. This could make Supplementary Table T3 obsolete and be more informative.

Fig. 2: Include rotation axes in panel. Pdb-codes/references of structures shown should be included in legend.

P12, L 300: heading should include 'purification' e.g. 'DARPin library, selection and purification' The discovery of the small molecule compound rigosertib to inhibit Ras-effector interactions (PMID: 27104980) should be included in the introduction or discussion.

Reviewer #2 (Remarks to the Author):

Using phage display and a designed ankaryrin repeat scaffold (DAPin) Guillard et al. report on two antibody mimetics that bind RAS in either the GTP- or GDP-bound state, designated K55 and K27, respectively. They characterize these engineered proteins at the biochemical and structural level and then show that, when expressed in cells, both bind RAS and inhibit MAPK signaling and oncogenic KRAS dependent growth in soft agar. The strength of the study is the thorough and rigorous biochemical characterization of the engineered proteins that include co-crystallization with RAS. The biological characterization is relatively underdeveloped as it applies to a single cancer cell line. Even were the cell based studies strengthened and perhaps extended in vivo to xenografts, the work would be more suitable for a specialized protein engineering journal than for Nature Communications because no new biology is revealed with the clever new reagents, the approach is not novel (see Ref. 11-13, 14, 31), and, most important, there is no pathway forward to a therapeutic since the reagents are genetically encoded macromolecules.

Specific comments:

The effects of DARPIn K27 and K55 on cell signaling (Fig. 5) are limited to cytofluorometric analysis of pErk in individual Hct116 cells. It would be informative to also study pMEK, pAKT, 2D growth in plastic, apoptosis and to use a variety of cell types including RAS dependent and independent tumor cell lines.

GFP-RAF1-RBD has been informative in showing where and when in cells RAS is activated (e.g. PMID 11988737, 15705808). It would be interesting to use GFP-K27 and GFP-K55 as probes for spatiotemporal RAS signaling in live cells.

K55 is predicted to function in a manner identical to the isolated RBD of a RAS effector such as RAF1. It would be informative to compare the two. Also, it is not possible to derive a cell line that stably expresses GFP-RAF1-RBD because the probe functions as a powerful dominant negative of the RAS/MAPK pathway required for cell proliferation and survival. Why is this not the case for K55?

The isolated CDC25 domain of RAS GEFs such as SOS bind strongly to the GDP-bound and the nucleotide-free transition state of RAS. However, presumably unlike K27, CDC25 promotes nucleotide exchange by weakening the affinity of the GTPase for both guanine nucleotides. It would be informative to compare CDC25 to K27.

The new compound ARS-853 (Ref. 17, 18) binds to GDP-loaded RAS-G12C and stabilizes the GDP-bound state. The prediction is that binding of this RAS isoform to K27, but not K55, would be dramatically enhanced by ARS-853.

Reviewer #3 (Remarks to the Author):

Small G-proteins cycle between a GDP and a GTP bound state. Interaction with effector proteins occur only in the GTP bound state. The exchange of GDP and GTP is catalysed by Guanine Nucleotide Exchange Factors (GEFs). The hydrolysis of GTP to GDP by the small G-protein is catalysed by GTPase activation proteins (GAPs). This process is impaired by cancer causing mutations in small G-proteins of the Ras family. Signalling by Ras proteins can be blocked by

competing with effector binding or by competing with the action of the GEFs.

Guillard et al. selected DARPins by phage display for their ability to bind to the oncogenic K-RasG12V mutant. Two DARPins are characterised in detail, one of which, K27, binds preferentially to the GDP bound state of Ras. K27 therefore competes with GEFs. The second DARPin, K55, binds preferentially to the GTP bound state of Ras. Crystal structure of the K27-Ras-GDP and the K55-Ras-GTP γ S complexes were solved. When expressed in HCT116 cells both DARPins inhibit EGF induced ERK phosphorylation and anchor independent growth.

The experiments are in general well performed and conclusive. The identification of a DARPin and thus of a protein, that binds selectively to the GDP bound form of Ras is of interest, as the naturally occurring effector proteins are selective to the GTP bound form. The obtained affinity of about 3 nM is excellent.

The authors discuss the therapeutic limitation of DARPins adequately.

Major points:

1. The characterisation of the interaction with the DARPins is of central importance, but the data are mainly presented in the supplement or not at all. Supplementary Figure 3 (Bio-layer interferometry) should be presented as main figure. Furthermore the figure should in addition include the data on the interaction between K27 and RasGDP, K55 and RasGTP, and K55 and RasGTP.

Data on the inhibition of Sos catalysed nucleotide exchange "MANT Assay" should be presented.

2. The obtained structural data (Fig1 and Fig2) are not presented clearly. For example, the overlay of K55 and RAF (Fig. 2b) is hardly "visible". The figures should be combined to one figure as this allows a better comparison of K27 and K55. Table 2 (Data collection and refinement statistic) can be moved to the supplement.

3. Monitoring of ERK-phosphorylation

Guillard et al. state "Since plasmid transfection is not efficient in all cells in a population, we choose to use a flow cytometry based assay in order to gate on cells which had been successfully transfected to express a FLAG-tagged DARPin, rather than performing Western blots for total phosphoERK (pERK) levels." It is neither explained in the material and methods section nor in the figure how gating was performed (Were cells only with high expression levels selected etc.).

Guillard et al. have generated cell lines with stable expression of K27 and K55 under a doxycycline inducible promoter. Western Blot for pERK should be performed in these cell lines and the data be included in Fig. 5.

Minor points:

1. The concept of DARPins should be introduced briefly, as this may not be common knowledge.

Reviewer #1 (Remarks to the Author):

The manuscript by Guillard et al. describes the biochemical, structural and cellular characterization of two designed ankyrin repeat proteins (DARPin) in their interaction with Ras proteins. Ras is a major driver in many oncogenic processes and has been a target for a number of biochemical approaches in order to inhibit downstream signaling of oncogenic Ras mutants but also considered undruggable for various reasons. The authors use phage display approaches followed by a Ras biochemical coupled assay to isolate two DARPins, termed K27 and K55, respectively. While K27 bound to GDP-bound RasG12V to inhibit Sos mediated nucleotide exchange, K55 is reported to act to inhibit the interaction between Ras and its downstream target Raf-kinase. The DARPins seem to act independently of the Ras-Isoform or mutant status as demonstrated by testing inhibition of N- and K-Ras along with several known oncogenic mutations. This is in contrast to previously reported compounds that depended on the

presence of the G12C-mutation. Nucleotide exchange activity seems blocked primarily by stabilization of the GDP-bound form, as has been reported for small molecule compounds specifically acting on the G12C mutant Ras. However, as mentioned above, in this work DARPins seem to act on several Ras isoforms and be independent of the mutational status of Ras. Crystal structures of Ras-DARPin complexes show the K27 binding site overlapping with that of guanine nucleotide exchange factor Sos and of K55 with that of Raf-RBDs. Intracellular interaction between Ras and DARPins was probed using a mammalian 2-hybrid system with a luciferase readout. Inhibition of Ras-Raf interaction in cells was not dependent on the presence of membrane localization motifs. To corroborate the notion that Ras-inhibition by DARPins affect Ras/MAPK signaling the authors demonstrate reduction of pERK levels after transfection with DARPin encoding plasmids and EGF stimulation of cells. Finally the authors

demonstrate DARPin mediated growth inhibition in soft agar colony formation assays.

In summary the paper presents a comprehensive biochemical, structural and cellular analysis of a novel designed Ras inhibitory protein scaffold. The structures of Ras-DARPin complexes define for the first time the interface between Ras and designed proteins that inhibit nucleotide exchange and act on a relatively broad variety of Ras variants or mutants. The research is important especially in light of a number of other studies, most importantly the small molecule Ras inhibitor rigosertib, after a long time of silence in the Ras-drugging field. Exploration of the molecular details of the interaction may enable design of chemical compounds/components that act by a similar mechanism and understand the mechanistic details of the interaction. The technical quality is high, results and methods are presented clearly, conclusions are justified, discussion appropriate.

I recommend publication of the paper in Nature Communications considering the following comments/suggestions:

P2: l 40ff: Original switch definition according to Milburn et al. (1990) is: switch I, res. 30-38; switch II, 60-76.

The original switch definitions have been inserted (p2, line 40) and the Milburn paper cited.

Regarding approaches to modulate Ras function, increase of GTPase activity may also be an approach for modulating Ras function that should be particularly valid in view of the observations that most oncogenic Ras mutant show impaired GTPase activity. Consider this aspect for introduction/discussion.

The concept of increasing the GTPase activity of oncogenic Ras mutants has been added to the list of options discussed for Ras modulation in the introduction (p2, line 42) and two relevant citations added to the Reference section.

P7: L148: Why was H-Ras used in the 2-hybrid assay and not K-Ras as for the in vitro interaction studies?

H-Ras was used in the 2-hybrid assay due to reagent availability and prior performance in the assay. Given the high homology of H-Ras to K-Ras, particularly in the regions bound by the two lead DARPins, we felt it was reasonable to use H-Ras in this assay, the primary intention of which was to assess DARPIn stability in the reducing environment of the cytoplasm. The high homology between K-Ras and H-Ras is now mentioned at an appropriate point in the Results section (p7, line 149).

Fig. 1: rotation axis and angle connecting panels in a) should be included in figure. I do not recognize RasGDP in purple. The DARPIn-Ras interface is seems of the most important features of the paper and should therefore be shown in more detail along with a cartoon specifying the interactions (e.g. similar like in Tanaka et al., 2007, ref.) for the biologist. This could make Supplementary Table T3 obsolete and be more informative.

The rotation axis has been added to Fig. 1 and a comment on the angle of rotation added to the legend. The Ras GDP structure in Fig. 1b has been re-coloured to blue in order to make it more distinct. An additional figure (Fig. 4 in the new figure numbering) has been added to better illustrate the interactions in the DARPIn-Ras interface for both DARPIn K27 and K55 (similar to the Tanaka et al., 2007 ref). A description of Fig. 4 has also been added to the Results section (p.6 lines 141-144). Supplementary Table 3 (now 4) has been retained as it shows the comparison to prior Ras-Raf and Ras-iDab6 structures, which does not feature in the new Fig. 4.

Fig. 2: Include rotation axes in panel. Pdb-codes/references of structures shown should be included in legend.

The rotation axis has been added and the PDB code included in the legend.

P12, L 300: heading should include 'purification' e.g. 'DARPIn library, selection and purification'

The suggested change has been made (p12, line 300)

The discovery of the small molecule compound rigosertib to inhibit Ras-effector interactions (PMID: 27104980) should be included in the introduction or discussion.

The introduction has been modified to include the small molecule Ras-mimetic rigosertib (p.3, lines 45-46) and the suggested citation has been added to the Reference section.

Reviewer #2 (Remarks to the Author):

Using phage display and a designed ankaryrin repeat scaffold (DAPin) Guillard et al. report on two antibody mimetics that bind RAS in either the GTP- or GDP-bound state, designated K55 and K27, respectively. They characterize these engineered proteins at the biochemical and structural level and then show that, when expressed in cells, both bind RAS and inhibit MAPK signaling and oncogenic KRAS dependent growth in soft agar. The strength of the study is the thorough and rigorous biochemical characterization of the engineered proteins that include co-crystallization with RAS. The biological characterization is relatively underdeveloped as it applies to a single cancer cell line. Even were the cell based studies strengthened and perhaps extended in vivo to xenografts, the work would be more suitable for a specialized protein engineering journal than for Nature Communications because no new biology is revealed with the clever new reagents, the approach is not novel (see Ref. 11-13, 14, 31), and, most important, there is no pathway forward to a therapeutic since the reagents are genetically encoded macromolecules.

The authors believe that the novelty of this study revolves around the discovery and characterisation of potent, new inhibitors of Ras nucleotide exchange, which is an exciting approach to tackling this important oncoprotein. In contrast to the previous work on

nucleotide exchange inhibition, we have isolated potent molecules which are active across all Ras isoforms and mutants tested. This lays the groundwork for ourselves and others to perform further studies which can answer additional questions relating to Ras biology which Reviewer #2 outlines below. Wherever possible within the scope of this manuscript, and as outlined in our point-by-point responses below, we have endeavoured to answer these questions to strengthen the biology aspects of this paper.

The references cited as reducing the novelty of our study are either describing Ras inhibitors with mechanisms of action completely unrelated to nucleotide exchange (Ref. 11-13, 31) or have not been sufficiently characterised for their cellular effects on Ras signalling (Ref. 14). The latter compound, DCAI (from Ref. 14), was also recently suggested to be acting via an off-target effect (see Ostrem and Shokat, Nature Reviews Drug Discovery 15, 771).

On the specific question of the path forward to therapeutic use of our molecules, the authors would like to point out that several valid paths forward have been suggested in the Discussion (p11, lines 279-281) and that these biochemical tools, which we describe for the first time in this publication, can also be useful in enhancing our understanding of Ras biology and enabling the discovery of therapeutics with the same mechanism of action.

The effects of DARPin K27 and K55 on cell signaling (Fig. 5) are limited to cytofluorometric analysis of pErk in individual Hct116 cells. It would be informative to also study pMEK, pAKT, 2D growth in plastic, apoptosis and to use a variety of cell types including RAS dependent and independent tumor cell lines.

In addition to cytofluorometric analysis of pERK we also show that DARPins K27 and K55 inhibit Ras binding to downstream effectors PI3K-alpha, PI3K-gamma, RALGDS and RAF1 in cells (Fig. 6) and cause a significant reduction in growth of the Ras mutant cell line HCT116 in a 3-D soft agar growth assay (Fig. 8). In response to the Reviewer's suggestion we have also included an additional Western blot experiment to consolidate the effect of K27 and K55 on ERK phosphorylation downstream of Ras (Fig 7a and additional text in Results, p 8, lines 187-196). In this experiment we also show that K27 and K55 are able to reduce phosphorylation of AKT (Fig 7a). We will follow up on proliferation and apoptosis effects across a broader range of tumour cell lines in future studies.

GFP-RAF1-RBD has been informative in showing where and when in cells RAS is activated (e.g. PMID 11988737, 15705808). It would be interesting to use GFP-K27 and GFP-K55 as probes for spatiotemporal RAS signaling in live cells.

As outlined above, the authors believe that an experiment on spatiotemporal profiling of Ras using these DARPins would be an interesting follow up study to this work but feel it is beyond the scope of this paper which describes the discovery of potent tools for the exploration of Ras nucleotide exchange inhibition.

K55 is predicted to function in a manner identical to the isolated RBD of a RAS effector such as RAF1. It would be informative to compare the two. Also, it is not possible to derive a cell line that stably expresses GFP-RAF1-RBD because the probe functions as a powerful dominant negative of the RAS/MAPK pathway required for cell proliferation and survival. Why is this not the case for K55?

This is a very interesting observation and agrees with some of our own data. We deliberately chose an inducible gene expression system to generate our stable cell lines because of concerns that uncontrolled expression of Ras-inhibitory DARPins would reduce cell proliferation and survival and lead to loss of the transgene. Despite using a regulated gene expression system we have observed, at later passages, a reduction in DARPin expression, particularly K55, which is in agreement with the results seen using GFP-

RAF1-RBD. As such we only show data from these cell lines where experiments were performed at an early passage.

The isolated CDC25 domain of RAS GEFs such as SOS bind strongly to the GDP-bound and the nucleotide-free transition state of RAS. However, presumably unlike K27, CDC25 promotes nucleotide exchange by weakening the affinity of the GTPase for both guanine nucleotides. It would be informative to compare CDC25 to K27.

The Ras/Sos structure (Boriack-Sjodin et al., (1998) Nature 394, 337) makes a striking contrast to our K27/Ras structure. In the Ras/Sos structure there is no nucleotide or magnesium bound to Ras and the nucleotide binding site is opened up by a drastic change in the switch 1 region. The distortion of switch 1 caused by Sos breaks the network of contacts with the nucleotide and therefore weakens the binding affinity for GDP and GTP. In contrast, the Ras in our structure has the GDP nucleotide present and as we show in the inset in Figure 1 (now Figure 2) the switch 1 loop can be superimposed on the switch 1 loop in the non-liganded GDP-Ras structure (PDB code: 4TQ9), suggesting a lack of distortion. The GDP in the K27 and non-liganded structures is also in a near-identical orientation. This suggests that K27 binds in a way which does not substantially modify the nucleotide pocket of Ras, in contrast to the very different mode of binding of Sos which completely disrupts the binding pocket, leading to nucleotide dissociation. The description of the K27/Ras structure in Results has been updated to make note of the comparison with Sos binding (p6, line 122-124).

The new compound ARS-853 (Ref. 17, 18) binds to GDP-loaded RAS-G12C and stabilizes the GDP-bound state. The prediction is that binding of this RAS isoform to K27, but not K55, would be dramatically enhanced by ARS-853.

The binding sites of ARS-853 and K27 appear to be non-overlapping from crystallographic studies, suggesting that simultaneous binding of both ARS-853 and K27 is possible, although there are some differences in the switch 2 conformation in the two structures. Since the binding of ARS-853 is covalent, the pre-incubation of GDP-loaded RAS-G12C with ARS-853 in vitro would be equivalent to simply adding GDP-loaded RAS in the absence of GTP and SOS, i.e. all RAS would be in the GDP form and therefore available for K27 binding. We would predict K27 to have the same binding affinity in the presence or absence of ARS-853 under these conditions. However, in a dynamic system such as in a G12C mutant cancer cell, where hydrolysis of the GTP form was also factored in, it would be of interest to look at the potential synergy between the two compounds and this could be a useful follow up in a future study to further characterise K27 activity.

Reviewer #3 (Remarks to the Author):

Small G-proteins cycle between a GDP and a GTP bound state. Interaction with effector proteins occur only in the GTP bound state. The exchange of GDP and GTP is catalysed by Guanine Nucleotide Exchange Factors (GEFs). The hydrolysis of GTP to GDP by the small G-protein is catalysed by GTPase activation proteins (GAPs). This process is impaired by cancer causing mutations in small G-proteins of the Ras family. Signalling by Ras proteins can be blocked by competing with effector binding or by competing with the action of the GEFs.

Guillard et al. selected DARPins by phage display for their ability to bind to the oncogenic K-RasG12V mutant. Two DARPins are characterised in detail, one of which, K27, binds preferentially to the GDP bound state of Ras. K27 therefore competes with GEFs. The second DARPIn, K55, binds preferentially to the GTP bound state of Ras. Crystal structure of the K27-Ras-GDP and the K55-Ras-GTP γ S complexes were solved. When expressed in HCT116 cells both DARPins inhibit EGF induced ERK phosphorylation and anchor

independent growth.

The experiments are in general well performed and conclusive. The identification of a DARPIn and thus of a protein, that binds selectively to the GDP bound form of Ras is of interest, as the naturally occurring effector proteins are selective to the GTP bound form. The obtained affinity of about 3 nM is excellent.

The authors discuss the therapeutic limitation of DARPins adequately.

Major points:

1. The characterisation of the interaction with the DARPins is of central importance, but the data are mainly presented in the supplement or not at all. Supplementary Figure 3 (Bio-layer interferometry) should be presented as main figure. Furthermore the figure should in addition include the data on the interaction between K27 and RasGDP, K55 and RasGTP, and K55 and RasGTP.

Data on the inhibition of Sos catalysed nucleotide exchange “MANT Assay” should be presented.

The bio-layer interferometry data for K27 and K55 on the GDP and GTP loaded forms of Ras has now been included as the first figure in the paper and the MANT assay data has also been included (Supplementary Fig. 4).

2. The obtained structural data (Fig1 and Fig2) are not presented clearly. For example, the overlay of K55 and RAF (Fig. 2b) is hardly “visible”. The figures should be combined to one figure as this allows a better comparison of K27 and K55. Table 2 (Data collection and refinement statistic) can be moved to the supplement.

In line with this request and that of Reviewer 1 we have separated the K55/Ras and Raf/Ras images to improve the visibility of the structures (Fig. 3b). We have also included an additional figure (Fig. 4) to more clearly represent and compare the Ras binding interfaces of K27 and K55. The data collection and refinement statistics have been moved to Supplementary Table 3.

3. Monitoring of ERK-phosphorylation

Guillard et al. state “Since plasmid transfection is not efficient in all cells in a population, we choose to use a flow cytometry based assay in order to gate on cells which had been successfully transfected to express a FLAG-tagged DARPIn, rather than performing Western blots for total phosphoERK (pERK) levels.” It is neither explained in the material and methods section nor in the figure how gating was performed (Were cells only with high expression levels selected etc.).

A sentence has been added to the Methods section to clarify that for cells transiently transfected with FLAG-tagged DARPins, cells were first gated for high FLAG expression prior to pERK analysis (p 16, line 411).

Guillard et al. have generated cell lines with stable expression of K27 and K55 under a doxycycline inducible promoter. Western Blot for pERK should be performed in these cell lines and the data be included in Fig. 5.

In order to provide Western blot verification of the flow cytometry data on pERK levels we have performed an additional experiment using transient co-transfection of cells with plasmids encoding (i) Ras and (ii) DARPins to demonstrate pERK knockdown (Fig 7a). In this experiment we also show that K27 and K55 are able to reduce phosphorylation of AKT (Fig 7a). Upon researching the literature on Ras-inhibitory antibodies, it appears that other groups have also demonstrated pERK / pAKT knockdowns in this way (e.g. Tanaka and Rabbits Oncogene. 2010 29:6064-70)

Minor points:

1. The concept of DARPins should be introduced briefly, as this may not be common knowledge.

Additional wording has been added to the Introduction to explain the relevant features of DARPins which make them well suited for this study (p 3, line 60-62)

REVIEWERS' COMMENTS:**Reviewer #1 (Remarks to the Author):**

The concerns I raised in the initial review have been appropriately addressed. I recommend publication of the MS in nature communications.

Reviewer #2 (Remarks to the Author):

The authors have responded to my comments and those of two other reviewers and have revised their manuscript accordingly. The response is thorough, scholarly and very clearly written. They have done an outstanding job with the revision. The one aspect that I wish could be further addressed is the narrow focus on a single Kras-dependent cell line (Hct116). Nevertheless, my major concern that the manuscript lacks novelty has been allayed. This is an interesting manuscript the is suitable for publication in Nature Communications.

Reviewer #3 (Remarks to the Author):

I am satisfied with the changes made